In situ observation of holopelagic Sargassum distribution and aggregation state across the entire North Atlantic from 2011 to 2020

http://orcid.org/0000-0003-0152-3009 Goodwin Deborah S. 1
http://orcid.org/0000-0002-4591-4063 Siuda Amy N.S. 2 siudaan@eckerd.edu
http://orcid.org/0000-0003-4180-6839 Schell Jeffrey M. 1
1 Sea Education Association , Woods Hole, Massachusetts , United States
2 Marine Science, Eckerd College , St. Petersburg, Florida , United States
Ragazzola Federica
Electronic publication date: 2022 Sep 22
Publication date: 2022
Volume: 10
Electronic Location ID: e14079
Received 2022 Jun 10; Accepted 2022 Aug 29
Copyright: © 2022 Goodwin et al.
Copyright year: 2022
Copyright holder: Goodwin et al.
License: This is an open access article distributed under the terms of the Creative Commons Attribution License, which permits unrestricted use, distribution, reproduction and adaptation in any medium and for any purpose provided that it is properly attributed. For attribution, the original author(s), title, publication source (PeerJ) and either DOI or URL of the article must be cited.
License URL: https://creativecommons.org/licenses/by/4.0/

Keywords: Windrow, Aggregation, In situ observations, Holopelagic Sargassum

Funding: Sea Education Association and Eckerd College Chair of Ocean Studies from the Henry L. & Grace Doherty Charitable Foundation This research was funded by the Sea Education Association and Eckerd College. Jeffrey M Schell was supported by a Chair of Ocean Studies from the Henry L. & Grace Doherty Charitable Foundation. There was no additional external funding received for this study. The funders had no role in study design, data collection and analysis, decision to publish, or preparation of the manuscript.

==============================
Between 2011 and 2020, 6,790 visual observations of holopelagic Sargassum were recorded across the North Atlantic Ocean to describe regional distribution, presence, and aggregation state at hourly and 10 km scales. Influences of oceanographic region and wind/sea conditions as well as temporal trends were considered; marine megafauna associates documented the ecological value of aggregations. Holopelagic Sargassum was present in 64% of observations from the western North Atlantic. Dispersed holopelagic Sargassum fragments and clumps were found in 97% of positive observations whereas aggregated windrows (37%) and mats (1%) were less common. Most field observations noted holopelagic Sargassum in quantities below the AFAI algorithm detection limit for the MODIS sensor. Aggregation state patterns were similar across regions; windrow proportion increased with higher wind speeds. In 8 of 10 years in the Sargasso Sea holopelagic Sargassum was found in over 65% of observations. In contrast, the Tropical Atlantic and Caribbean Sea exhibited greater inter-annual variability (1–88% and 11–78% presence, respectively) that did not align with extremes in central Atlantic holopelagic Sargassum areal coverage determined from satellite observations. Megafauna association patterns varied by taxonomic group. While some study regions were impacted by holopelagic Sargassum dynamics in the equatorial Atlantic, the Sargasso Sea had consistently high presence and operated independently. Field observations capture important dynamics occurring at fine spatiotemporal scales, including transient aggregation processes and ecological value for megafauna associates, and therefore remain essential to future studies of holopelagic Sargassum.

Introduction

Holopelagic Sargassum, a brown macroalgae, drifts at the water’s surface creating a unique and ecologically-significant floating marine ecosystem in the oligotrophic open ocean (Haney, 1986; Wells & Rooker, 2004). Presently, two species and multiple morphological forms of holopelagic Sargassum are recognized and distributed throughout the equatorial and western North Atlantic Ocean (Parr, 1939; Butler et al., 1983; Goodwin, Siuda & Schell, 2020; Dibner et al., 2021). At the <1 to 100s of meters spatial scale, holopelagic Sargassum aggregation patterns underpin its ecological value, the dynamic availability of highly localized prey biomass within and beneath the algae, and remote sensing detection and prediction capabilities (Moser & Lee, 2012; Wang & Hu, 2017; Wang et al., 2019).

Holopelagic Sargassum may be observed in multiple states (Fig. 1; Parr, 1939; Butler et al., 1983; Marmorino et al., 2011; Ody et al., 2019). Fragments are short fronds less than 20 cm in length, broken off a clump. Individual clumps may be roughly spherical or oblong in shape when afloat, with 20 cm or greater frond length or clump diameter. Both fragments and clumps are stand-alone units of holopelagic Sargassum and can be found dispersed or aggregated (Butler et al., 1983; Ody et al., 2019). Windrows are defined as aggregations of many fragments and/or clumps in a line generally arranged parallel to the wind direction (Figs. 1B and 1C); they may extend for multiple km distance and range in width from <0.5 m to several m (Woodcock, 1950; Marmorino et al., 2011). Mats are densely-packed holopelagic Sargassum fragments and/or clumps with a clear edge, irregular to round shape, and 5 to 100s m distance across. Often mats are observed associated with a windrow, exhibiting noticeably greater diameter/width (4–5 times) compared to the average of the windrow itself (Figs. 1D and 1E; Marmorino et al., 2011; Ody et al., 2019); under calm conditions, mats may be observed as completely independent features not connected to a windrow (Parr, 1939).

Figure 1 Holopelagic Sargassum offshore aggregation states.

(A) Dispersed clumps and fragments (0.2 m diameter clumps; Caribbean, winter 2021, credit J. Schell). (B) Narrow windrow (0.5–1.5 m across; Tropical Atlantic, fall 2015, credit A. Siuda). (C) Wide windrow (3–4 m across; Caribbean, winter 2015, credit J. Schell). (D) Small mats (3 × 9 m mat near small boat; Sargasso Sea, fall 2014, credit C. Morrall). (E) Large mats (50 × 100 m; near Guadeloupe, Caribbean, spring 2015, credit J. Schell).

The processes controlling formation and dissipation of holopelagic Sargassum windrows and mats as well as the timescale of transitions between local aggregation states remain poorly understood. Among early observations of abundant windrows, Parr (1939) noted large mats appeared during periods of low wind, suggesting increased aggregation under calm conditions. However, Woodcock (1950) documented holopelagic Sargassum windrows at wind speeds from 1 to 17 m s–1. Though transient, wind-induced helical vortices (Langmuir cells) can reach many km in length, are positively correlated in both horizontal and vertical dimensions with greater wind speeds, and persist as cohesive features for substantial periods of time under consistent winds (Langmuir, 1938; Faller & Woodcock, 1964). Reanalyzing Parr’s (1939) dataset, Woodcock (1993) found that surface density of holopelagic Sargassum decreased above moderate (>4 m s−1) wind speeds, such that any windrows present appeared less well-defined due to strong vertical mixing. After wind changes direction or abates, windrows reorient or dissipate on the order of 2 to 30 min (Leibovich, 1983). Recent analysis of spectrographic and infrared images collected during repeat plane transects over a single hour showed signs of floating Sargassum mats disintegrating as wind or waves pushed the main portion of the mat downwind faster than upwind edges, causing a trail of less aggregated vegetation (Marmorino et al., 2011).

While the physical process of Langmuir Circulation results in surface windrows across a range of wind speeds, the local abundance of holopelagic Sargassum will determine the observed level of aggregation. For example, Butler et al. (1983) most frequently observed holopelagic Sargassum as dispersed clumps arranged in rows separated by 10s of m and aligned with the wind. At low abundance, individual clumps and fragments may appear dispersed when they are actually arranged in a diffuse windrow.

Holopelagic Sargassum serves as critical nursery, spawning, foraging, roosting, and protective habitat for a diversity of invertebrate, fish, sea turtle, marine mammal, and seabird species (Dooley, 1972; Morris & Mogelberg, 1973; Martin et al., 2021). Smaller taxa reside within the architecturally-complex branches of clumps and fragments while larger and/or migratory species are associated with patchy accumulations on a transient or long-term basis (Moser, Auster & Bichy, 1998; Wells & Rooker, 2004; Moser & Lee, 2012; Martin et al., 2021). Aggregated mats were biodiversity hotspots near Suriname, South America, supporting significantly higher abundances of juvenile sea turtles, cetaceans, and foraging seabirds than neighboring open waters (de Boer & Saulino, 2020). Likewise, at western North Atlantic locations seabird density was over 30 times greater where floating Sargassum was present, findings amplified near large mats (Haney, 1986); foraging birds utilized a range of strategies to capture their invertebrate and fish prey (Moser & Lee, 2012). Size and longevity of aggregations impacted the diversity of associated fishes, with far more taxa observed under mats (19) than clumps (12) or absent Sargassum (four; Moser, Auster & Bichy, 1998). Four species of sea turtles (loggerhead, green, hawksbill, and Kemp’s ridley) exhibited strong dependence on holopelagic Sargassum at the early-juvenile stage, during which the macroalgal mats provided shelter as well as ready access to their primary diet of associated invertebrate fauna (Witherington, Hirama & Hardy, 2012). The ecological role of holopelagic Sargassum is particularly important in nutrient-poor subtropical and tropical waters (Laffoley et al., 2011; de Boer & Saulino, 2020).

Since 2011, annually recurring holopelagic Sargassum coastal inundation events across the equatorial Atlantic have resulted in negative economic and ecological impacts (van Tussenbroek et al., 2017; Bartlett & Elmer, 2021; Hendy et al., 2021; Oxenford et al., 2021; Maurer et al., 2021), thus prompting advances in satellite detection and transport models for prediction of beaching times and locations. Originally, the Maximum Chlorophyll Index (MCI; only available for MERIS and OLCI sensors) was used in remote detection of floating Sargassum (Gower et al., 2006; Gower & King, 2011), but this index does not differentiate macroalgae from intense phytoplankton blooms. The more recent Alternative Floating Algal Index (AFAI; Wang & Hu, 2016; Wang et al., 2019), designed for all ocean color satellite sensors (MODIS, VIIRS, MERIS, OLCI), specifically detects floating macroalgae based on the enhanced red-edge reflectance around 750 nm. Because of the high sensitivity of the MODIS instrument, floating Sargassum may be detected with only 0.2% (2,000 m2) overall coverage of a 1 km2 pixel (Wang & Hu, 2016). Using near real-time AFAI images and the Floating Algal density product distributed via the Sargassum Watch System (SaWS; Hu et al., 2016), the trajectories of large holopelagic Sargassum aggregations have been modeled to predict beaching events (Marechal, Hellio & Hu., 2017; Marsh et al., 2021; Trinanes et al., 2021). However, after applying the AFAI algorithm and neural network deep learning to high-resolution sensor data, Wang & Hu (2021) concluded that floating Sargassum density is greatly underestimated using MODIS, the magnitude to which has not been ground-truthed. Clearly, field observations continue to provide valuable insight supplementary to satellite data; for example, minimal floating macroalgae of limited distribution was detected in the Caribbean by Gower & King (2011) and the Sargasso Sea by Wang et al. (2019), when shipboard surveys found persistent holopelagic Sargassum in those regions during the same periods (Goodwin, Siuda & Schell, 2020).

Initially motivated to complement Gower & King’s (2008, 2011) satellite-derived floating Sargassum distribution cycle, the field surveys presented here cover the full North Atlantic and are highly spatiotemporally resolved (hourly and every 10 km). This work leveraged unparalleled in situ observations to describe holopelagic Sargassum presence and aggregation state over 10 years and at spatial scales below those detectable using remote sensing. Influences of oceanographic region and wind/sea conditions on holopelagic Sargassum distribution and aggregation state were considered. Regional inter-annual patterns of holopelagic Sargassum were compared to better understand their connectivity and dynamics of the equatorial Atlantic. The relationship between temporal variability of satellite-detected areal coverage and in situ observations of holopelagic Sargassum was also determined to describe the predictive capabilities and detection limits of remote sensing tools. Marine megafauna associates documented the ecological value of holopelagic Sargassum aggregations.

Materials and Methods

Field protocols

Between 2011 and 2020, 6,790 “hourly observations” were recorded onboard Sea Education Association’s SSV Corwith Cramer during most daylight hours (Fig. 2). Observations from 61 oceanographic research cruises across the North Atlantic Ocean and Caribbean Sea were included in this analysis, 38 6-week and 23 10–14-day cruises. All were cruises of opportunity; the voyage tracks were not designed specifically for monitoring holopelagic Sargassum distribution, but rather to optimize weather and sailing conditions throughout the year. Cruise tracks varied seasonally and annually (Fig. 2). In most years, the vessel followed an annual cycle departing Massachusetts south-bound in October, sailing in the western tropical Atlantic and Caribbean from November through April, and transiting north through the Sargasso Sea in May. From autumn 2014 through spring 2017, the vessel followed an annual cycle that included a northern trans-Atlantic crossing from Massachusetts to the Mediterranean (June to October), a southern trans-Atlantic track arriving in the Caribbean in mid-December, winter sailing in the Caribbean, and the north-bound transit through the Sargasso Sea in May.

Figure 2 Hourly observation locations within eight sampling regions.

Light blue circles indicate where holopelagic Sargassum was absent, navy circles where it was present. Dotted lines are borders between oceanographic regions. Sampling months for each region noted at lower right. GoM, Gulf of Mexico; FS, Florida Straits; Shelf, New England Shelf.

Each daytime observation occurred near the top of the hour for a 6-min period (1/10th of an hour). Part of a larger research effort tracking seabirds, marine mammals, sea turtles, fish, marine debris, and holopelagic Sargassum, hourly observations were systematically recorded regardless of macroalgal presence. Standing on the Cramer’s quarterdeck at approximately 4 m height of eye, observers documented the presence and quantity of any megafauna within the entire field of view and any isolated target floaters (debris, holopelagic Sargassum, other vegetation) alongside the port side of the ship. Holopelagic Sargassum was recorded in multiple categories reflecting its general abundance in the region and different states of aggregation: fragments, clumps, windrows, and mats. Fragments and clumps were counted when within 5 m of the ship. Windrows were counted when they passed perpendicular to the observer or within 50 m of the ship. Large mats were recorded when seen, typically within 100 m of the ship but occasionally at greater distance during calm conditions. Over 99.9% of hourly observations of clumps and windrows included a count; fragments were described as present/absent. From the deck of the ship, no taxonomic differentiation of Sargassum spp. was possible. However, detached and floating benthic Sargassum was rarely encountered while quantifying each Sargassum species in twice-daily surface net tows conducted during the same offshore cruises (Schell, Goodwin & Siuda, 2015; Goodwin, Siuda & Schell, 2020).

The date, time, GPS position, Beaufort Force, visibility, and relevant notes about key sightings were captured as metadata with each hourly observation. Beaufort Force (BF, from 0–12) relates wind speed and sea surface conditions, thus reflecting the physical processes impacting local floating Sargassum transport better than wind speed alone. All data points were assigned to one of eight regions defined on the basis of physical oceanographic boundaries (Fig. 2).

Data analysis

Quality control confirmed accurate, consistent data entry and GPS positions, and filtered out incomplete records; 131 hourly observations were excluded in this process and the remaining 6,659 analyzed. Wind speeds were categorized into low (BF 0–2), moderate (BF 3–5), and high (BF 6–9). Holopelagic Sargassum aggregation states were ranked from small dispersed pieces to cohesive structures with defined boundaries (fragments/clumps < windrows < mats); each observation was assigned the highest aggregation state recorded during that period. Because the Eastern Atlantic and New England Shelf regions included a low proportion (2% and 21%, respectively) of observations with holopelagic Sargassum present, they were excluded from a subset of analyses. All statistics were performed in JMP Version 16 (SAS Institute Inc., Cary, NC, USA).

Holopelagic Sargassum presence/absence counts were summarized as percentages across geographic regions, wind/sea conditions, aggregation types, and years. Fisher’s exact tests were used to compare proportions between regions or conditions. For temporal analyses in the Tropical Atlantic, Caribbean Sea (merged east and west), and Sargasso Sea (merged north and south), data were averaged on an ‘annual’ scale to integrate across the spatio-temporal covaried nature of the cruise tracks. Years were here defined as the period from October in the previous year through May in the current year. June through September observations were primarily on the New England Shelf, outside the typical range of holopelagic Sargassum, and thus were excluded from these annual means. Pearson’s product-moment correlation analyses of the ‘annually’ averaged presence/absence and aggregation states of holopelagic Sargassum were performed among the three larger regions to assess temporal coherence of observed patterns.

The linear relationships between annual holopelagic Sargassum presence, windrow, and mat aggregation frequencies from in situ observations, and satellite-derived areal coverage were explored with regression analysis. October through May monthly mean remote sensing areal coverage data for a zone encompassing the present hourly observation dataset were used to calculate annual mean satellite-derived areal coverage estimates. Monthly mean data were provided by the Optical Oceanography Laboratory at the University of South Florida (https://optics.marine.usf.edu), using the same approach as in Wang & Hu (2016) and Wang et al. (2019). Years with mean satellite-derived areal coverage less than 500 km2 were defined as low coverage, based on a natural break in the bimodal distribution.

Noted megafauna species were grouped into seabirds, marine mammals, sea turtles, and fish; mammal and turtle categories were not analyzed due to low numbers of encounters. Fisher’s exact tests were used to compare percent megafauna presence between observations grouped by presence or absence of holopelagic Sargassum and, when the floating macroalgae was present, grouped by aggregation state.

Results

Holopelagic Sargassum was present in 3,076 (46%) of 6,659 total observations distributed across eight regions (Fig. 2). Dispersed fragments and clumps were most common (in 97% of positive holopelagic Sargassum observations). In contrast, aggregated windrows and mats were recorded in 37% and 1% of positive holopelagic Sargassum observations, respectively. When holopelagic Sargassum was present, only 1% of observations included all three aggregation states, 34% of observations included two aggregation states (99% of those were windrows concurrent with dispersed fragments and clumps), and 65% of observations included a single aggregation state (96% of those were dispersed fragments and clumps).

Counts of dispersed clumps and windrows seen during each observation period provided additional context to the dataset. Counts were recorded for 99.9% of the 2,264 dispersed clump observations and for all of the 1,137 windrow observations. When present, 1–5 individual clumps were most common; only 11% of observations included more than 25 clumps (Fig. 3A). Records including 10 or fewer windrows accounted for 85% of the positive observations (Fig. 3B).

Figure 3 Holopelagic Sargassum counts.

Number of observations within each count range for dispersed (A) clumps and (B) windrows. Observations without pelagic holopelagic Sargassum not shown.

With regard to observed wind conditions and sea states, BF ranged from 0–9 across the dataset. 75% of observations were conducted in moderate conditions (BF 3–5), while only 8% were conducted at high BF and 17% at low BF. A similar pattern of observation frequency across BF categories resulted when the dataset was parsed by region; 67–91% of the observations were conducted in moderate conditions (Table 1). Holopelagic Sargassum was present in 38% of low BF observations, 48% of moderate BF observations, and 48% of high BF observations.

Table 1 Observations as a function of Beaufort force (BF).

	Tropical Atlantic	East	West	GoM/FL Strait	South	North	Shelf	Eastern Atlantic	
	Caribbean Sea	Sargasso Sea	
nTotal	465	991	388	208	1,453	848	818	1,350	
Low, BF 0–2	6%	14%	19%	21%	15%	16%	21%	23%	
Moderate, BF 3–5	91%	81%	72%	71%	79%	73%	70%	67%	
High, BF 6–9	3%	5%	9%	8%	6%	11%	9%	10%	
Note:

Percent of observations in each of three BF categories, reflecting combined wind conditions and sea state. GoM, Gulf of Mexico; FL, Florida.

The proportion of observations with holopelagic Sargassum present was near 50% or greater in each region, except for the New England Shelf and the Eastern Atlantic (Table 2). Proportions present in these two regions were significantly different from each other (p < 0.0001) and significantly lower than proportions present in all other regions (p < 0.0001 for each pairwise comparison). The overall present portion increased from 46% to 64% when the New England Shelf and the Eastern Atlantic were excluded from the calculation. Consequently, the remaining analyses focused on data from the six regions where holopelagic Sargassum was most frequently observed.

Table 2 Observations as a function of region.

	Tropical Atlantic	East	West	GoM/FL Strait	South	North	Shelf	Eastern Atlantic	
	Caribbean Sea	Sargasso Sea	
nTotal	467	1,050	398	212	1,491	856	829	1,356	
sig.	a,b	a	b	c	c,d	d	e	f	
Present	53%	48%	58%	68%	74%	75%	21%	2%	
Absent	47%	52%	42%	32%	26%	25%	79%	98%	
Note:

Percent of observations during which holopelagic Sargassum was present or absent in each oceanographic region. GoM, Gulf of Mexico; FL, Florida. Letters identify statistically significant differences between regions based on Fisher’s exact test results.

Across all regions, holopelagic Sargassum was present only as dispersed fragments and clumps in 57–73% of observations and windrows, whether alone or accompanied by dispersed fragments and clumps, were present in 25–41% of observations (Fig. 4A). Mats were not seen in the Tropical Atlantic or Western Caribbean and were rarely documented in other regions (Fig. 4A). The proportion of windrows was lowest at BF 1–3 (range: 23–33%) and greatest at BF 4–6 (range: 40–43%; Fig. 4B). All pairwise comparisons for each of the low BF windrow proportions against each of the high BF proportions were significantly different (p-values < 0.03), whereas all pairwise comparisons of windrow proportions within low and high BFs were not significantly different (p-values > 0.05). Mats were observed at all wind conditions, except for BF 7 (Fig. 4B).

Figure 4 Holopelagic Sargassum aggregation by region and Beaufort Force.

Proportion of highest ranked aggregation states by (A) region and (B) BF. As only one positive holopelagic Sargassum observation each was made at BF 0 and 8, these levels not shown; holopelagic Sargassum was absent from the observation at BF 9. GoM, Gulf of Mexico; FL, Florida.

Temporal patterns of holopelagic Sargassum presence and aggregation state were explored for three larger oceanographic regions (Fig. 5). The Sargasso Sea exhibited consistently high percent presence over the examined decade, with holopelagic Sargassum recorded in greater than 65% of observations in 8 of 10 years; the minimum, in 2020, was still 48% (Fig. 5C). In contrast, the Tropical Atlantic and Caribbean Sea exhibited greater inter-annual variability in the proportion of in situ observations with holopelagic Sargassum present (Figs. 5A and 5B). Annual holopelagic Sargassum presence between the Caribbean Sea and Tropical Atlantic was positively correlated (r = 0.91, df = 8, p = 0.0019), whereas correlations between those regions and the Sargasso Sea were not significant (r values < 0.18, df = 9, p-values > 0.50). In most years in all regions, dispersed clumps and fragments alone were more commonly observed than aggregated holopelagic Sargassum (Figs. 5D–5F). Nonetheless, windrows were observed annually in the Caribbean Sea and Sargasso Sea. With the exception of years with low total percent presence in the Tropical Atlantic and Caribbean, windrows were recorded in greater than 30% (and frequently in greater than 50%) of observations (Figs. 5D–5F). A positive relationship, though not as strong as for presence, was observed for temporal patterns in windrow occurrence between the Caribbean Sea and Tropical Atlantic (r = 0.53, df = 8, p = 0.18), whereas correlations with the Sargasso Sea were again not significant (r values < 0.09, df = 9, p-values > 0.50). Temporal correlation in mat occurrence between the Caribbean Sea and Sargasso Sea was not significant (r = −0.11, df = 9, p = 0.78), and no mats were observed in the Tropical Atlantic.

Figure 5 Annual patterns of holopelagic Sargassum presence and aggregation type.

(A–C) Annual (October through May) proportional presence and (D–F) ranked aggregation states in the Tropical Atlantic, merged Eastern and Western Caribbean regions, and merged North and South Sargasso Sea regions. Low annual mean satellite-derived areal coverage (<500 km2) of holopelagic Sargassum in the central Atlantic Ocean indicated by underscores; all other years were high coverage. Remote sensing data as monthly means, derived using the same approach as in Wang & Hu (2016) and Wang et al. (2019), were provided by the Optical Oceanography Laboratory at the University of South Florida (https://optics.marine.usf.edu).

Annual patterns of in situ observed holopelagic Sargassum presence, windrows, and mats were also related to remotely-sensed areal coverage across the three larger oceanographic regions (Fig. 5). The only significant trend observed was a positive linear relationship between Caribbean occurrence of windrows and areal coverage (R2 = 0.69, p = 0.006, n = 9; Fig. 5E). Otherwise, total presence, windrows, and mats did not increase in proportion or frequency during high areal coverage years in the Tropical Atlantic (R2 values < 0.06, p-values > 0.558, n = 8) or Sargasso Sea (R2 values < 0.10, p-values > 0.400, n = 9; Figs. 5A–5D and 5F). During 2013, the lowest satellite-derived coverage year, holopelagic Sargassum presence in the Tropical Atlantic and Caribbean Sea was also low, 28% and 36%, respectively, but 76% for the Sargasso Sea. In 2018, when areal extent was greatest, holopelagic Sargassum was present in only 47% of observations in the Caribbean Sea and 78% in the Sargasso Sea; field surveys were not conducted in the Tropical Atlantic that year.

Megafauna count was recorded during 99.7% of total hourly observations. A total of 4,459 fish, 1,633 seabirds, 551 marine mammals, and 12 sea turtles were documented, indicating 37% overall megafauna presence in the dataset (Table 3). Marine mammal and sea turtle observations were too infrequent to reveal patterns of association with holopelagic Sargassum. Fish were noted more frequently when holopelagic Sargassum was present and when aggregated (Table 3); significant differences were observed with presence of holopelagic Sargassum in the Tropical Atlantic (p = 0.0003) and with aggregation in the Sargasso Sea (p = 0.0139). Seabirds exhibited the opposite pattern; they were present more frequently when holopelagic Sargassum was absent (Table 3), with significant differences observed in the Tropical Atlantic (p = 0.0027) and the Caribbean Sea (p = 0.0004). Where seabirds were observed concurrent with holopelagic Sargassum, they were more frequently associated with dispersed clumps and fragments than windrows and mats (Table 3), although the increases were not statistically significant in any region.

Table 3 Observations of megafauna.

Holopelagic Sargassum	Fish presence	Seabirds presence	
TA	CS	SS	TA	CS	SS	
Absent	27.3%	13.6%	14.9%	16.8%	27.6%	11.0%	
Present	43.3%	15.2%	14.2%	7.7%	19.6%	10.7%	
Clumps & Fragments	41.2%	15.2%	12.4%	10.1%	21.5%	11.3%	
Windrows & Mats	46.5%	15.3%	16.8%	4.0%	17.0%	9.8%	
Note:

Percent presence for fish or seabirds as a function of holopelagic Sargassum presence/absence (top rows) across all hourly observations and as a function of ranked aggregation state (bottom rows) for the subset of observations when holopelagic Sargassum was present. Italic/bold indicates significant differences p-values < 0.05 in percent megafauna presence between categories. TA, Tropical Atlantic; CS, Caribbean Sea (merged Eastern and Western Caribbean regions); SS, Sargasso Sea (merged North and South Sargasso Sea regions).

Discussion

Over the centuries, explorers, mariners, and oceanographers have attempted to map the distribution of holopelagic Sargassum (Ardron et al., 2011). Though each drew a slightly different map, a general pattern emerged and thus the central waters of the North Atlantic gyre became known as the Sargasso Sea. However, beginning in 2011 this historic precedent was overturned with the first holopelagic Sargassum inundation event across the tropical North Atlantic (Franks et al., 2012). For nearly a decade since, the authors of the present study have documented with field observations the occurrence and aggregation states of holopelagic Sargassum across a majority of the basin. The resulting unprecedented dataset demonstrates that the epipelagic waters of the tropical North Atlantic have experienced a dramatic and lasting transformation.

Holopelagic Sargassum is now a recurring feature of open ocean pelagic environments well beyond the historic boundaries of the Sargasso Sea, extending into the equatorial Atlantic, Caribbean Sea, and Gulf of Mexico (Schell, Goodwin & Siuda, 2015; Torres-Conde, 2022) where it is referred to as the Great Atlantic Sargassum Belt (GASB; Wang et al., 2019). Though the Sargasso Sea may have ‘seeded’ the tropical Atlantic with holopelagic Sargassum in 2010–11 (Johns et al., 2020), the 10 years of field observations presented here demonstrate that the GASB is sustained independently. Critically, as the GASB has waxed and waned over the examined decade, holopelagic Sargassum was consistently present throughout the western North Atlantic (North and South Sargasso Sea regions; Figs. 2 and 5). In notable contrast, holopelagic Sargassum was nearly absent in this project’s Eastern Atlantic region throughout the entire study period (Fig. 2; Table 2). The nature of prevailing winds and currents across the tropical Atlantic and the rarity of holopelagic Sargassum in the Eastern Atlantic region suggests an emergent source of holopelagic Sargassum in the equatorial Atlantic supplying the GASB, a conclusion supported by several modeling efforts (Franks, Johnson & Ko, 2016; Brooks et al., 2018; Johnson et al., 2020). Floating Sargassum transport models to date have solely focused on expansive mats and associated buoyancy, size, and windage estimates (Putnam et al., 2018; Johns et al., 2020; Berline et al., 2020; Marsh et al., 2021); however, if most holopelagic Sargassum is not aggregated in mats (Figs. 4 and 5D–5F) and parameterizations rarely reflect the actual size and shape of clumps or mats (Miron et al., 2020; Putnam et al., 2020; van Sebille et al., 2021), the current suite of model assumptions may be incomplete.

Occurrences of holopelagic Sargassum aggregated in extensive mats spanning 10s of m2 or more were rare in this dataset (Fig. 4A). Initially, these findings appear to contradict the persistent story that the North Atlantic is home to vast expanses of holopelagic Sargassum that could impede the passage of sailing vessels, a story dating back to the early accounts of Columbus’ voyages (Keen, 1959) and echoed throughout history (Maury, 1857). Yet, for centuries other historical records have described holopelagic Sargassum accumulation in more modest terms (Beebe, 1926), some going so far as to suggest that the more fanciful accounts of ‘Sargassum prairies’ stretching to the horizon were mere hyperbole (Gordon, 1941). Far more commonly, this project observed holopelagic Sargassum as dispersed clumps and fragments or windrows (Figs. 4A and 5D–5F). The recurring holopelagic Sargassum inundation events that began in 2011 on windward beaches and harbors across west Africa, the Caribbean Sea, and the Gulf of Mexico have indeed reached mythic proportions (Smetacek & Zingone, 2013; Partlow & Martinez, 2015; Hu et al., 2016; van Tussenbroek et al., 2017). In some instances, however, inundation events may be the result of holopelagic Sargassum build-up arriving as dispersed clumps and windrows.

Ody et al. (2019) examined holopelagic Sargassum aggregation patterns across the Tropical Atlantic and Eastern Caribbean regions during two cruises in 2017. They more frequently observed mats of holopelagic Sargassum (their Type 4 and 5 rafts, 24% of observations) than the present study. However, Ody et al.’s (2019) station locations were non-random, as their cruises were routed using satellite images and model simulations targeting areas of high floating macroalgal concentration. As their study only reported the highest ranked Sargassum raft type observed at each station, direct comparisons were not possible.

Though vast expanses of aggregated holopelagic Sargassum were rare across the broad spatial and temporal range of this study, a few locations reliably harbored mats. On the leeward sides of large islands in the Eastern Caribbean (e.g., Dominica and Guadeloupe) and in the Windward Passage a combination of local currents, winds, and bathymetry produced substantial aggregations of holopelagic Sargassum, including mats and large windrows (Fig. 1). Significantly and similarly, Ody et al. (2019) observed the majority of their large holopelagic Sargassum mats in two specific locations: south of Cape Verde Islands and leeward of Guadalupe. Research cruise design for the present study did not permit a rigorous examination of such phenomena at local scales. However, the ecological and fisheries implications of large, persistent aggregations of holopelagic Sargassum on the leeward sides of Caribbean and other islands is worthy of future investigation.

In the open ocean, dispersed clumps and fragments were more abundant than windrows across all BFs and regions (Figs. 4A and 4B). At higher BFs (4–6), the proportion of windrow observations increased significantly (Fig. 4B); nonetheless, under high BF conditions aggregations may have been broken apart by larger waves or more difficult to discern as algal material was carried below the sea surface (Woodcock, 1993). Ody et al. (2019) did not observe a similar relationship between holopelagic Sargassum aggregation state and wind speeds, findings likely influenced by their limited sample size (n = 42) and narrower range of conditions (BF 2–6). As BFs were similar in all studied regions (Table 1), geographic position and prevailing weather (e.g., the trade winds in the Tropical Atlantic or the mild environment of the Sargasso Sea) did not appear to drive observed aggregation states. Rather, the instantaneous physical forces underlying Langmuir Circulation (Leibovich, 1983) and holopelagic Sargassum abundance in any given location determined windrow aggregation. This effort did not record or examine wind direction and duration, which may have been influential to the holopelagic Sargassum states observed. Windrows noted at low BF (Fig. 4B) may have, at times, documented Langmuir-driven features in the process of forming or dissipating as wind speed or direction shifted (Langmuir, 1938; Leibovich, 1983). Hourly observations on the SSV Corwith Cramer were typically made while the vessel was underway at speeds <6 knots, meaning each record captured a snapshot in space and time of holopelagic Sargassum presence and aggregation state, BF, and megafauna associates.

Greater numbers of fishes were observed with holopelagic Sargassum present relative to absent, and greater numbers near aggregated windrows and mats than dispersed clumps and fragments (Table 3). Past studies found similar patterns, with enhanced abundance and species diversity of fishes amongst floating Sargassum compared to open water; significant relationships between fish quantity and algal wet weight have also been noted, though aggregation states were not recorded (Wells & Rooker, 2004; Casazza & Ross, 2008). Moser, Auster & Bichy (1998) described mat morphology (size of continuous floating Sargassum) and availability duration as influential to associated fish diversity. Small fishes live within and immediately proximate to holopelagic Sargassum of any density, relying upon the architecturally-complex habitat for protection, associated epibiont and motile epifauna prey resources, and as a nursery. Larger predators school deeper beneath windrows and mats to capitalize on congregated prey (Moser, Auster & Bichy, 1998).

Seabirds associating with holopelagic Sargassum were recorded more frequently around disaggregated fragments and clumps than near dense windrows and mats, though were present under both conditions (Table 3). de Boer & Saulino (2020) likewise reported a non-significant difference in seabird abundance between holopelagic Sargassum presence/absence, while Haney (1986) noted significantly higher bird densities around mats. Although not examined to such detail in this data, in previous work some seabird species (tropicbirds, boobies, terns) exhibited a preference for larger windrows and mats to target flying fish prey and employ plunge-diving and aerial-dipping foraging strategies (Haney, 1986; Moser & Lee, 2012). Smaller taxa and those picking invertebrate and macrofaunal prey from within the holopelagic Sargassum (phalaropes, shearwaters) were more common near small patches and dispersed clumps. When comparing surveys in loose vs densely aggregated holopelagic Sargassum, the former attracted larger groups of foraging seabirds as suitable prey was more visible and accessible (de Boer & Saulino, 2020). Different holopelagic Sargassum morphotypes host motile invertebrate communities of varied abundance, diversity, and value as a food resource (Martin et al., 2021); hourly observations reported here did not attempt to identify holopelagic Sargassum clumps or aggregations to morphotype, but this detail may inform interpretation of future megafauna studies.

While the ecological value of holopelagic Sargassum scales with clump and aggregation size (Stoner & Greening, 1984; Martin et al., 2021), even dispersed fragments and clumps offer important niches within the open ocean environment. In this geographically and temporally expansive dataset, dispersed clumps and windrows were far more common than mats; both provide excellent habitat for mobile megafauna regardless of their ever-changing, wind-driven aggregation status. Species requiring persistent aggregated holopelagic Sargassum for survival at certain life stages (e.g., juvenile sea turtles; Witherington, Hirama & Hardy, 2012) may be challenged by the dominance of dispersed clumps. As aggregation is intricately tied to algal abundance, in years and/or regions with lower overall holopelagic Sargassum biomass (Figs. 5A–5C), large mats may never form or be sustained because insufficient material is available on a local scale. Extremely dense macroalgal aggregations, such as the vast quantities washing into Caribbean bays from the GASB, can have negative impacts to coastal ecosystems (reduced light penetration, lowered seawater dissolved oxygen levels, decaying material; Chavez et al., 2020; Hendy et al., 2021). Whether the same ecologically detrimental effects occur offshore beneath massive mats is as yet unstudied.

With the desire to predict and prepare for potentially catastrophic beaching events, communities have come to rely upon bulletins based on satellite-derived estimates of floating Sargassum distribution and abundance (Hu et al., 2016; Marechal, Hellio & Hu., 2017; Trinanes et al., 2021). AFAI strength is used to assign fractional coverage estimates of floating Sargassum to remote sensing pixels for algorithm outputs. Therefore, it is valuable to consider what satellites actually detect given the majority of this project’s hourly observations did not contain aggregated windrows or mats. Wang & Hu (2016) estimated that for a 1 km2 MODIS pixel only 0.2% algal coverage (2,000 m2) was necessary for floating Sargassum detection using the AFAI algorithm. For comparison with this lower limit, at an average five knot ship speed the distance covered during one 6-min hourly observation period reported here is 926 m, approximately equivalent to the length of a pixel. Dispersed clumps and fragments were accurately quantified within a 5 m wide swath next to the ship, thus a single observation period represents ~0.5% of a MODIS pixel. Based on a generous assumption of 25 (Fig. 3A) dispersed 0.3 m diameter clumps per observation period, up-scaling from the present study yields estimated holopelagic Sargassum clump areal coverage of only 354 m2, well below Wang & Hu’s (2016) detection limit. If assumed to extend for a full km perpendicular to the ship’s track, only 2 1-m wide windrows of densely packed holopelagic Sargassum would be required to achieve the lower detection limit. Windrows were found in only 30–40% of observations (Fig. 4), and it was rare to encounter densely packed windrows (Figs. 1 and 3B), therefore a greater quantity of narrow or diffuse windrows in close proximity would be needed to reach 2,000 m2 coverage for satellite detection. Based on this exercise, it is not surprising that Rodriguez-Martinez, Jordan-Dahlgren & Hu (2022) reported a mismatch between highest monthly mean quantities of holopelagic Sargassum collected from beaches along the northern Mexican Caribbean coast and those derived from satellite observations of localized regions of interest in adjacent waters. Likewise, Torres-Conde (2022) conclude that the lack of floating macroalgae reported in remote sensing products covering the northwest coast of Cuba did not coincide with the low (relative other Caribbean locations), but measurable, holopelagic Sargassum accumulations on the beach.

Over the study period, the authors observed holopelagic Sargassum in a variety of growth stages, from large healthy clumps to dead fragments. In the Tropical Atlantic during November and December 2014, holopelagic Sargassum clumps were robust, bright in color, with many branches and visible new growth, indicating an ongoing bloom (Figs. 6A and 6B). In contrast, in 2015 windrows were frequently present but most contained decaying darkened clumps or broken apart fragments, pneumatocysts, and blades (Figs. 6C and 6D); these aggregations of dead material were likely the remains of the preceding extensive GASB. As floating Sargassum was detected by AFAI in both cases (Wang et al., 2019), growth stages may not be differentiated. In practice, remote sensing provides a synoptic view that cannot be matched in geographic scale by field observations. The AFAI algorithm applied to MODIS images provides a good approximation of holopelagic Sargassum distribution and abundance for areas and times when drifting macroalgae is abundant and concentrated. However, dispersed holopelagic Sargassum clumps as well as narrow and/or diffuse windrows are undetectable on most occurrences at coarse (1 km2) resolution (Wang & Hu, 2021) and satellite-derived biomass may not always reflect an active macroalgal bloom.

Figure 6 Holopelagic Sargassum at different growth stages.

(A) Windrow of healthy holopelagic Sargassum clumps (~35 cm diameter clumps; Tropical Atlantic, fall 2014, credit J. Schell). (B) Healthy clump of Sargassum natans VIII (30 cm ruler shown; Tropical Atlantic, fall 2014, credit J. Schell). (C) Windrow of decaying fragments and pneumatocysts (~0.7 m wide; Tropical Atlantic, fall 2015, credit A. Siuda). (D) Decaying clump of Sargassum natans VIII (1 cm grid shown; Tropical Atlantic, fall 2015, credit A. Siuda).

Annual patterns of holopelagic Sargassum presence and aggregation in the Tropical Atlantic and Caribbean Sea generally did not exhibit a relationship with GASB areal coverage determined through satellite image analysis (Fig. 5). Thus, when remote sensing characterized a weak GASB (Wang et al., 2019), high percent presence of holopelagic Sargassum fragments, clumps, and even windrows were noted in the hourly observations presented here. High inter-annual variability in the Tropical Atlantic and Caribbean Sea may be driven by local wind and current dynamics, bathymetry, and differences in cruise track from 1 year to the next. Significantly, the Sargasso Sea remained relatively stable with respect to holopelagic Sargassum percent presence and windrow frequency (Figs. 5C, 5F) over all studied years regardless of GASB fluctuations.

The implications of the reported results are three-fold. First, dispersed clumps dominate offshore holopelagic Sargassum presence substantially over windrows and mats. Second, coarse resolution remote sensing is unable to resolve ecologically-meaningful occurrences of dispersed or small-scale aggregated holopelagic Sargassum compared to in situ observations. Third, the Sargasso Sea appears to be operating independently from the equatorial Atlantic and dynamics of the GASB. Consistent holopelagic Sargassum presence since 2011 demonstrates that the tropical North Atlantic has entered a new regime, worthy of ongoing investigation into its inter-annual dynamics and ecology as well as, potentially, newly defined boundaries for the “Sargasso Sea.”

Supplemental Information

Supplemental Information 1 Raw data.

All dates, local times, GPS positions, wind conditions, as well as numbers of each aggregation state and associated fauna for each observation.

Click here for additional data file.

We thank the students, crew, and faculty of SEA Semester (Sea Education Association, Woods Hole, MA USA) cruises on the SSV Corwith Cramer since 2011 for making detailed hourly observations. This manuscript benefited from valuable comments from three reviewers.

Additional Information and Declarations

Competing Interests

Author Contributions

Data Availability

The authors declare that they have no competing interests.

Deborah S. Goodwin conceived and designed the experiments, performed the experiments, analyzed the data, prepared figures and/or tables, authored or reviewed drafts of the article, and approved the final draft.

Amy N. S. Siuda conceived and designed the experiments, performed the experiments, analyzed the data, prepared figures and/or tables, authored or reviewed drafts of the article, and approved the final draft.

Jeffrey M. Schell conceived and designed the experiments, performed the experiments, analyzed the data, prepared figures and/or tables, authored or reviewed drafts of the article, and approved the final draft.

The following information was supplied regarding data availability:

The raw data are available in the Supplemental File.

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
