# Peer review of "In situ observation of holopelagic Sargassum distribution and aggregation state across the entire North Atlantic from 2011 to 2020"

_PeerJ, doi:10.7717/peerj.14079_

## Round 0.1 · original submission · Major Revisions

Dear authors, all Reviewers agreed that the data you provided are very important for the scientific community.

However, more attention needs to be paid in the material and methods section. In particular, you should improve the clarity of the experimental design and describe which statistical analysis you performed. For example, in the results, there are statistical test results that are not mentioned in the data analysis section.

·

Basic reporting

There are some problems with the literature cited
1- Be consistent in placing the doi in the references. Examples: Bartlett, D., Elmer, F., 2021. The impact of Sargassum inundations on the Turks and Caicos Islands. Phycol. 1, 83-104. doi:10.3390/phycology1020007.

Brooks, M.T., Coles, V.J., Hood, R.R., Gower, J.F.R., 2018. Factors controlling the seasonal
459 distribution of pelagic Sargassum. Mar. Ecol. Prog. Ser. 599, 1-18.
460 https://doi.org/10.3354/meps12646.

2- Place bibliographic references of the data taken from https://optics.marine.usf.edu.
Example: SaWS., 2022. Satellite-based Sargassum Watch System. Optical Oceanography Laboratory. The University of South Florida. Data retrieved on April 20, 2022, from: https://optics.marine.usf.edu/projects/SaWS/pdf/Sargassum_outlook_2021_bulletin11_USF.pdf.

3- I suggest discussing your results with the paper by Torres-Conde (2022) "Is simultaneous arrival of pelagic Sargassum and Physalia physalis a new threat to the Atlantic coasts?"


*Change the font of the letters in Figure 2. It can't read well what it says.

Experimental design

*The methods need to be better explained for better understanding.

1- In Figure 5 there is a comparison of field observation data with remote sensing data (from the site https://optics.marine.usf.edu). However, there is no mention in the methods section of where the remote sensing data is taken from. I suggest placing the source of the data you are comparing in the methods section.

2- In the discussion section on lines 415-418 are put statistical test results that are not mentioned in the data analysis section (i.e., R-squared values all < 0.103 and p-values all > 0.400). In my opinion, the result of this test should go in the results section and not in the discussion section. Additionally, the name of this statistical test that you used should be placed in the data analysis section and for what purpose you use it.

3- In the data analysis section, it is stated that a Fisher's test was performed however in the results section the result of the test is not provided to allow evidencing the comparison of megafauna associated with pelagic Sargassum.

4- In general, I recommend the use of more statistical tests to strengthen the comparisons made in the results and to support the explanations given in the discussion section. I recommend some Chi-square tests or some variant of ANOVA transforming the data from discrete variable (counts) to continuous.

5- Please indicate the program where you performed all statistical analyses and its version.

Validity of the findings

*Adding statistical tests can help to strengthen the discussion of the results and make a better validation of the conclusions.

* I suggest discussing your results with the paper by Torres-Conde (2022) "Is simultaneous arrival of pelagic Sargassum and Physalia physalis a new threat to the Atlantic coasts?". In this paper, pelagic Sargassum sightings until 2022 are placed throughout the Atlantic Ocean. These authors found similar results: “In the western Atlantic Ocean more pelagic Sargassum sightings have been found than in the eastern Atlantic Ocean”. Additionally, the results of Torres-Conde (2022) also agree with the results of the present study in that the remote sensing data provided by the site https://optics.marine.usf.edu did not detect some pelagic Sargassum arrival events observed in situ. I believe that the comparison of the results of this study with those obtained by the work of Torres-Conde (2022) can strengthen the validation of the conclusions of this study.

·

Basic reporting

General comments
This manuscript is a valuable, timely and well-written contribution. The scope of the manuscript is appropriate for this journal. The data collected need to be shared with the scientific community. I support the publication of this manuscript. I provide a series of constructive comments; I hope authors, that I have a full respect for their careers and expertise, will read my comments knowing I value very much this manuscript. I submit my comments open and with full transparency regarding my concerns and opinions. My goal is to help improve the way we report the knowledge we are building up in relation of the issue of Sargassum in the Atlantic.
While English is professional and ideas are clear, I have some concerns with the way the genus name is used in the manuscript. This paper will provide important information, however, the background in terms of species and genus name use needs revision.
Some members of the community working on the Sargasso/ Sargassum issue are using the term Sargassum, which is a taxonomic descriptor as a common name. The publication of papers addressing the problem of Sargasso/ Sargassum influxes is growing; we need to be careful and consistent with nomenclatural rules and uses.
I do think that we need to follow nomenclatural rules on the proper use of a genus name in a non-taxonomical context, and use italics ONLY in a taxonomical context. A common name that is derived from a genus name, such as the gorilla, is not capitalized or italicized. The use of italics, not only by the authors of this paper but in several previous studies, is reducing the understanding of the diversity of the genus Sargassum which includes more than 350 species, and the fact the other species of the genus can have pelagic forms, not holopelagic but pelagic such as S. hornerii.

For example on line 39, the opening sentence: Pelagic Sargassum, a brown macroalgae, should be written as “Pelagic species of the genus Sargassum…….” Or simple pelagic sargassum, without italics, can be used as a non-taxonomic descriptor or common name

Another example in line 49 “Drifting Sargassum……” authors are using the genus name to refer to the specific problem of Sargasso. Drifting Sargassum in reality includes all species of the genus that can present drifting forms, more than the two holopelagic species can drift. The use of the genus name is misleading here too.

I suggest authors to provide a paragraph explicitly mentioning the fact that the genus Sargassum has many species, two of them known as holopelagic forming the Sargasso Sea. Clarifying what they mean/refer to when using Sargassum and I highly recommend using the italics only when using the word as a taxonomic descriptor, not as a common name. I do think this is a matter of historical use of the common name in a wrong way, the fact that previous authors have done that should not be continued. We are in a perfect moment to clarify and use taxonomic names as such, with italics, and common names as such without italics.

The use of the term drifting is similar/synonymous with pelagic? The use of the term drift can be confusing in algal papers. I suggest clarifying as well, the holopelagic Sargassum species drift/float? Some algae can drift in the bottom of coastal lagoons, for example, Mats of Laurencia species drift on seagrass beds. My suggestion is to use floating fragments that drift depending on winds etc…..

I suggest reducing significantly the section by analyzing all the problems related to satellite detection. In the introduction.

Line 263 Can you explain what you mean by tipping-point, are you using the word in terms of loss of resilience and moving to a new stage? I think you could use the idea as a conclusion, or provide more explanation. The series of observations without correlation to other parameters, from my perspective, is not sufficient to demonstrate a loss of resilience and a change to a new stable state if that is the context authors are using the term tipping-point. The tipping point is for the North Atlantic, or the tropical Atlantic including the equatorial Atlantic. I am not convinced by your discussion and observations that the large influxes affecting the Caribbean are coming from the Sargasso Sea in the North Atlantic, but rather from the Great Sargassum Belt. If that is the case the tipping point is for the Atlantic. Just as you discuss the exact following paragraph.
You retake this conclusion at the very end, it might be important to define what you understand by North Atlantic, in figure 2 it is not clear what you recognize as North Atlantic.
Finally, your conclusion of the tipping point comes after mentioning in line 422 that “Significantly, the Sargasso Sea remained relatively stable with respect to pelagic Sargassum percent presence and windrow frequency (Fig. 5C, F) over all studied years regardless of GASB fluctuations.” If the Sargasso Sea is stable and the Sargasso Sea is in the North Atlantic, the changes are actually in the Caribbean and Equatorial areas where the GASB is localized. I suggest clarifying your major discussion and supporting your major conclusion with statistical tests that demonstrate such stability, and significant differences among your data surveys, not only percentage observations are needed to demonstrate changes, either increase, decrease correlation or stability.


Line 353 versus should be in italics is a Latin word.

I really think this paper needs to be published, a few changes might bring the quality of your discussion and conclusions to a higher level.
I hope you find my comments useful.
Ligia

Experimental design

This manuscript report data from a survey is not an experiment. However, all surveys and monitoring programs do have a monitoring plan based on a statistical approach.

The data, while valuable and useful, lacks a description of the design process for the cruise tracks. The only explanation provided is that the tracks varied seasonally and annually but there are no explanations if the cruise track was randomly designed or followed a particular objective.

No statistical methods are described either; the data analysis section is a description of the quality control of collected data, and an explanation of what data were used, but not how they were used (correlations? probability tests?). The results are descriptive percentages of the different observations, not statistical analysis testing any hypothesis. However, in the discussion, some relationships analysis (lines 413-424) are discussed, but not mentioned in the results. It is not clear to me why the relationships discussed are not mentioned in the methods and results section. I suggest revisiting this section and include what statistical analyses were done.
Finally, the supplement data are codes in an Excel file, but I was not able to look at the raw data.

Validity of the findings

The data from the survey presented as percentages and bar graphs are very important and serve to ground-proof several oceanographic models.
However, they do not provide a full picture, and I think that recognizing the limitations, due to the survey design, and spatial and temporal extension, need to be addressed when concluding their support for Johnson et al model and discarding the Johns et al model.

Additional comments

I applaud the effort to put together the amount of data in a publication.

·

Basic reporting

Language is clear.
Introduction and background are fully relevant, although some references may be added for completeness
Structure is conform to standard.
Figures and labels are clear, but some may be improved.
The raw data are provided.

Detailed comments
L124 Wang & Hu (2021) did not simply applied AFAI to HR sensors but developed NN based detection based on FAI. They do conclude that Sargassum density was 30% higher with HR sensors than using MODIS, but it is not constant across scenes and there is no ground truth.
Fig 1 light blue not visible
Fig 4 : add confidence interval
Fig 5 : presence/absence: redundant
table 1 and 2 : percent of presence (avoid redundance)

Experimental design

Questions are well defined, investigation and methods are generally ok.

Detailed comments
L143. The time distribution of the cruises is a key information that should be illustrated in the figures.
L205-210. Each time that qualitative variables are compared (contingency tables), a chi2 test should be carried out. For Fig 4, and tables 1, 2, 3.
L243: high areal coverage years: how do you define these?

Validity of the findings

L274-278: Johns et al (2020) suggested that the initiation of the new Sargassum region (GASB) was caused by a high wind event that translated into a large flux of Sargassum from the Sargasso Sea to the tropical Atlantic. They do not mean that the interannual variability of the GASB was controlled by a flux from the Sargasso sea.
Typo: Putman et al
Windage was in particular examined by Berline et al 2020 https://doi.org/10.1016/j.marpolbul.2020.111431.
L280-283. This is a good point, but authors need to clarify what are the hypothesis of the modeling: Basically all models use satellite detections mainly from MODIS, for input/validation, which means that they are only focusing on the aggregations, not on clumps.
L285-L299. You should compare your results to Ody et al 2019.
L380 “are used to code remote sensing pixels” Unclear. Basically Sargassum detection, at the pixel level, are the inputs of these models (initial position for Lagrangian particles).
L413 I think that the annual time scale is not appropriate to compare your dataset to remote sensing estimates, given the extremely high time and space variability of the aggregation. I suggest you change at least to monthly estimates if you have the data available.
L418 Is this sentence correct? “Weak GASB, high percent presence of fragments etc”.
L427: Be accurate: coarse resolution (1 km ) remote sensing detection. High resolution may overcome this limit.
L430 “not exchanging significant quantities”. How can you support this assertion?

Additional comments

Discussion may be improved by more focusing on the results.

---

## Round 0.2 · accepted · Accept

After carefully reading the rebuttal letter and checking the manuscript, I am pleased to inform you that your paper has been accepted. Thank you for adding a better description of the statistical analysis.